# AKR1B1 Represses Glioma Cell Proliferation through p38 MAPK-Mediated Bcl-2/BAX/Caspase-3 Apoptotic Signaling Pathways

Yu-Kai Huang [1,2,3] 🔾, Kun-Che Chang [1,4,5] 🔾, Chia-Yang Li [1], Ann-Shung Lieu [1,2] and Chih-Lung Lin [1,2,*]

[1] Graduate Institute of Medicine, College of Medicine, Kaohsiung Medical University, Kaohsiung 80708, Taiwan; yukaih@gmail.com (Y.-K.H.); kcchang@pitt.edu (K.-C.C.)
[2] Department of Neurosurgery, Kaohsiung Medical University Hospital, Kaohsiung 80708, Taiwan
[3] Division of Neurosurgery, Department of Surgery, Kaohsiung Municipal Ta-Tung Hospital, Kaohsiung Medical University, Kaohsiung 80145, Taiwan
[4] Department of Ophthalmology, Louis J. Fox Center for Vision Restoration, University of Pittsburgh School of Medicine, Pittsburgh, PA 15213, USA
[5] Department of Neurobiology, Center of Neuroscience, University of Pittsburgh School of Medicine, Pittsburgh, PA 15213, USA
* Correspondence: hnf4ahnf4a@gmail.com or u109800009@kmu.edu.tw

**Abstract:** This study aimed to investigate the regulatory role of Aldo-keto reductase family 1 member B1 (AKR1B1) in glioma cell proliferation through p38 MAPK activation to control Bcl-2/BAX/caspase-3 apoptosis signaling. AKR1B1 expression was quantified in normal human astrocytes, glioblastoma multiforme (GBM) cell lines, and normal tissues by using quantitative real-time polymerase chain reaction. The effects of AKR1B1 overexpression or knockdown and those of AKR1B1-induced p38 MAPK phosphorylation and a p38 MAPK inhibitor (SB203580) on glioma cell proliferation were determined using an MTT assay and Western blot, respectively. Furthermore, the AKR1B1 effect on BAX and Bcl-2 expression was examined in real-time by Western blot. A luminescence detection reagent was also utilized to identify the effect of AKR1B1 on caspase-3/7 activity. The early and late stages of AKR1B1-induced apoptosis were assessed by performing Annexin V-FITC/PI double-staining assays. AKR1B1 expression was significantly downregulated in glioma tissues and GBM cell lines (T98G and 8401). Glioma cell proliferation was inhibited by AKR1B1 overexpression but was slightly increased by AKR1B1 knockdown. Additionally, AKR1B1-induced p38 MAPK phosphorylation and SB203580 reversed AKR1B1's inhibitory effect on glioma cell proliferation. AKR1B1 overexpression also inhibited Bcl-2 expression but increased BAX expression, whereas treatment with SB203580 reversed this phenomenon. Furthermore, AKR1B1 induced caspase-3/7 activity. The induction of early and late apoptosis by AKR1B1 was confirmed using an Annexin V-FITC/PI double-staining assay. In conclusion, AKR1B1 regulated glioma cell proliferation through the involvement of p38 MAPK-induced BAX/Bcl-2/caspase-3 apoptosis signaling. Therefore, AKR1B1 may serve as a new therapeutic target for glioma therapy development.

**Keywords:** AKR1B1; glioblastoma; p38 MAPK; BAX; Bcl-2; caspase-3

## 1. Introduction

Gliomas are the most common brain tumors among adults [1]. Its most aggressive subtype is glioblastoma multiforme (GBM), which accounts for approximately 80% of malignant gliomas, and the median survival time is only 12–14 months [2]. Considering its diffuse and infiltrative characteristics, GBM exhibits outstanding abilities in cell proliferation, invasion, and migration, thereby limiting the scope for surgical removal and the effects of clinical treatments [3,4]. Elucidating the cellular and molecular mechanisms leading to GBM proliferation and invasion would shed light on new strategies for therapeutic intervention. Gliomas are a collection of brain tumors that arise from glial cells in the

central nervous system (CNS). Glioblastoma, also referred to as glioblastoma multiforme (GBM), represents the most aggressive and malignant subtype of gliomas, classified as grade IV glioma according to the World Health Organization (WHO) classification system [5]. In this manuscript, the terms glioma and glioblastoma are used interchangeably to specifically denote grade IV gliomas, characterized by swift growth, high invasiveness, and unfavorable prognosis.

Aldo-keto reductase family 1 member B1 (AKR1B1), a specific member of the Aldo keto reductase superfamily, catalyzes the reduction of glucose to sorbitol and regulates the polyol pathway of glucose and lipid metabolism [6]. AKR1B1 is involved in a complex network of signaling pathways, including oxidative stress, inflammation, and epithelial–mesenchymal transition [7]. Downregulating AKR1B1 reportedly leads to poor prognosis and significantly impacts survival in patients with pheochromocytoma and paraganglioma tumors [8]. AKR1B1 downregulation is also reported in hepatocellular carcinoma [9], endometrial cancer [10], sporadic adrenocortical tumors [11], and prostate cancer [6]. In addition, AKR1B1 has been implicated in the development of several human cancers and resistance to certain anticancer drugs [12–15]. However, the role of AKR1B1 in brain cancer remains unclear, but growing evidence suggests that it has a large impact on cancer therapy.

Bcl-2 and BAX are two important members of the Bcl-2 family of proteins that play critical roles in regulating programmed cell death or apoptosis. Bcl-2 is an anti-apoptotic protein that inhibits apoptosis, while BAX is a pro-apoptotic protein that promotes it [16,17]. In cancer cells, Bcl-2 is often overexpressed, allowing the cells to evade apoptosis and continue to proliferate. Conversely, reduced expression of BAX in cancer cells can also contribute to resistance to apoptosis [18–20]. Understanding the balance between Bcl-2 and BAX expression is crucial for developing targeted cancer therapies that can restore normal apoptotic regulation and induce cancer cell death [21–23].

Caspase-3 is a critical mediator of the apoptosis pathway, and its activity is reduced in glioma cells, contributing to their resistance to apoptosis and promoting tumor growth [24]. The balance between pro-apoptotic BAX and anti-apoptotic Bcl-2 proteins is known to regulate caspase-3 activity and apoptosis in glioma cells. Bcl-2 is an anti-apoptotic protein that inhibits caspase-3 activation and promotes cell survival, while BAX is a pro-apoptotic protein that promotes caspase-3 activation and apoptosis [25,26]. The dysregulation of Bcl-2 and BAX can lead to an imbalance that decreases caspase-3 activity and contributes to resistance against apoptosis. Targeting Bcl-2 and/or activating BAX to restore caspase-3 activity may be a potential therapeutic strategy for glioma [26,27]. Several strategies, such as small molecule inhibitors, gene therapy, and immunotherapy, have been explored to modulate BAX and Bcl-2 expression and restore caspase-3 activity in glioma cells, with promising results in preclinical studies [28–30].

Mitogen-activated protein kinases (MAPK) are serine/threonine protein kinases that are classified into three main subgroups: extracellular signal-regulated kinase 1/2, c-Jun amino (N)-terminal kinase (JNK), and p38 MAPK [31]. These kinases are activated in response to extracellular stimuli or biological stress and modulate a series of signal transduction in physiological processes [32]. In particular, p38 MAPK signaling pathways play pivotal roles in cell apoptosis [33,34]. In lung cancer and colon cancer, p38 MAPK phosphorylation induces apoptosis [35,36]. Phosphorylated p38 MAPK can promote apoptosis by downregulating Bcl-2 and upregulating BAX [37–40]. Studies have shown that phosphorylation of p38 MAPK can regulate caspase-3 activity in different cell types, including glioma cells [41,42]. Phosphorylation of p38 MAPK can activate caspase-3 and promote apoptosis [42]. Targeting p38 MAPK phosphorylation and its downstream signaling pathways may represent a potential therapeutic strategy for glioma by modulating caspase-3 activity and inducing apoptosis [43]. Apoptosis is a highly regulated process of programmed cell death that plays a critical role in maintaining organismal homeostasis [44]. During apoptosis, there is a loss of plasma membrane integrity, and phosphatidylserine, which is typically confined to the inner leaflet of the plasma membrane, becomes exposed on the outer surface. Moreover, caspase-3 functions as a central mediator of the apoptotic

pathway. Upon activation by internal or external stimuli, caspase-3 cleaves and activates downstream apoptotic proteins, culminating in the disassembly of cellular structures and functions. Therefore, the quantification of caspase-3 activity is a crucial means of evaluating the extent of apoptosis [44,45].

In this study, we aimed to investigate the roles of AKR1B1 in glioma cell proliferation and related mechanisms. We hypothesized that the AKR1B1 reduces glioma cell proliferation and activates p38 MAPK phosphorylation, thereby mediating the Bcl-2/BAX/caspase-3 pathway.

## 2. Materials and Methods

### 2.1. Materials

Cell-culture-related reagents were purchased from GIBCO-BRL (Grand Island, NY, USA). Fetal bovine serum (FBS) was purchased from HyClone (Logan City, UT, USA). Cell media were obtained from Thermo Scientific (Waltham, MA, USA). AKR1B1-entry plasmids, control plasmids, AKR1B1-targeting siRNAs, and nontargeting si-RNA control were ordered from OriGene (Rockville, MD, USA). Furthermore, DNA and si-RNA transfection reagents were purchased from jetPRIME (VWR, Radnor, PA, USA). An SYBR® Green PCR Master Mix and a MultiScribe™ Reverse Transcriptase Kit were obtained from Applied Biosystems (Foster City, CA, USA). Antibodies against AKR1B1, p-p38 MAPK, p38 MAPK, p53, Bcl-2, BAX, and beta-actin were obtained from Proteintech (Rosemont, IL, USA). A p38 MAPK inhibitor SB203580 was obtained from SmithKline Beecham Pharmaceuticals (King of Prussia, PA, USA). A caspase-Glo 3/7 Assay System was obtained from Promega (Annandale, NSW, Australia). An Annexin-V FITC/PI Apoptosis Kit was purchased from Elabscience Biotechnology Co., Ltd., (Houston, TX, USA). Finally, the MTT (3-(4,5-dimethylthiazol-2-yl)-2,5-diphenyltetrazolium bromide) reagent and all other chemicals were purchased from Sigma-Aldrich (St. Louis, MO, USA).

### 2.2. Patient Samples

All tissue samples used in this study were collected from patients undergoing surgical resection at Kaohsiung Medical University Hospital (Kaohsiung, Taiwan). Prior to the procedure, written informed consent was obtained from each participant. This study was approved by the Clinical Research Ethics Committee of Kaohsiung Medical University Hospital (KMUHIRB-G(II)-20170010). During glioma resection surgery, normal brain tissue was collected. The initial step of the procedure involved corticotomy, during which some normal brain tissue was removed. Consequently, normal brain tissue could be collected during the necessary steps of tumor resection. The specific location of the normal brain tissue was dependent on the tumor location, such as the frontal, parietal, or occipital lobe.

### 2.3. Cell Cultures

Normal human astrocytes (SV-1) and glioblastoma cell lines (U87, 8401, G5T, DBTRG-05MG, and T98G) were purchased from the American Type Culture Collection (Manassas, VA, USA). SV-1, U87, G5T, DBTRG-05MG (05MG), and T98G cells were cultured in an Eagle-Alpha modification solution supplemented with 10% FBS. A total of 8401 cells were grown in RPMI 1640 containing 10% FBS. All cells were incubated at 37 °C in a humidified atmosphere with 5% $CO_2$.

### 2.4. DNA Transfection and RNA Interference

A pCMV6-Entry vector, pCMV6-Myc-DDK-tagged-AKR1B1, or si-RNA was added into a jetPRIME buffer and then mixed gently with a jetPRIME transfection reagent according to the manufacturer's instructions. We incubated the transfection mixture for 10 min at room temperature and added it to cells in a growth medium. These cells were then incubated at 37 °C, and the transfection medium was replaced with a cell growth medium after 24 h transfection.

### 2.5. Caspase3/7 Activity Assay

After seeding the cells in 96-well plates at a density of $1 \times 10^4$ cells/well, we added 20 μL of Caspase-Glo 3/7 reagent to individual wells and control wells (growth media only). Subsequently, the plates were shaken at 300 rpm for 30 s and further incubated in the dark at 37 °C for 30 min. Luminescence was then measured using a microplate reader (Promega Corporation, Madison, WI, USA).

### 2.6. MTT Assay

For this assay, cells were also seeded in 96-well plates at a density of $1 \times 10^4$ cells/well. The viability of cells transfected with plasmids was measured at 0, 24, 48, 72, and 96 h, and that of cells treated with si-RNA was measured at 0, 1, 2, 3, 4, and 5 d. We added the MTT reagent (100 μL) to each well and incubated the cells for 4 h at 37 °C. After incubation, we carefully removed the MTT reagent and added 100 μL of dimethyl sulfoxide to solubilize the violet crystals. Next, the plates were shaken for 20 min in the dark to allow for complete solubilization. The optical density was measured at 490 nm using a microplate (ELISA) reader (Thermo Fischer, Waltham, MA, USA).

### 2.7. Quantitative Real-Time Polymerase Chain Reaction (qPCR)

Using the Trizol reagent, we extracted total RNA from GBM tissues or cultured cells. RNA was reverse-transcribed to cDNA using a MultiScribe™ Reverse Transcriptase Kit. We conducted qPCR by using a Real-Time PCR Detection system (Bio-Rad, Hercules, CA, USA) with an SYBR Green Master Mix according to the manufacturer's instructions. Glyceraldehyde 3-phosphate dehydrogenase was used as the internal control. The relative gene expressed as a fold-change was calculated using the $2^{-\Delta\Delta Ct}$ equation. The primers were as follows: AKR1B1 (forward: 5′- TTTTCCCATTGGATGAGTCGG-3′; reverse: 5′-CCTGGAGATGGTTGAAGTTGG-3′); BAX (forward: 5′-CCCGAGAGGTCTTTTTCCGA-3′; reverse: 5′-CCAGCCCATGATGGTTCTGAT-3′); Bcl-2 (forward: 5′-GGTGGGGTCATGTGTGTGG-3′; reverse: 5′-CGGTTCAGGTACTCAGTCATCC-3′); GAPDH (forward: 5′-GTGAAGGTCGGAGTCAAC-3′; reverse: 5′-GTTGAGGTCAATGAAGGG-3′).

### 2.8. Western Blot Analysis

Cells were harvested and lysed using RIPA buffer supplemented with protease inhibitors. The protein concentration in the resulting lysate was measured using a Bradford assay (Bio-Rad, Hercules, CA, USA). Total proteins were separated using 10% sodium dodecyl sulfate-polyacrylamide gel electrophoresis and transferred onto polyvinylidene difluoride membranes. For 1 h at room temperature, the membranes were blocked in phosphate-buffered saline with 0.1% Tween 20 containing 5% nonfat milk powder. Next, we incubated these membranes with primary antibodies overnight at 4 °C, followed by incubation with appropriate secondary antibodies for 1 h at room temperature. The chemiluminescence signal was visualized using an enhanced chemiluminescence detection kit system (PerkinElmer, Shelton, CT, USA).

### 2.9. Annexin V-FITC/PI Double-Staining Assay

A total of $8 \times 10^4$ cells were plated in a 12-well culture plate, and after 24 h, they were transfected with either 0.2 μg of a control plasmid or a plasmid expressing AKR1B1. Following 24 h of transfection, cells were gently washed with $1\times$ PBS to remove the excess medium. In accordance with the manufacturer's recommendations, an appropriate volume of Annexin V-FITC staining solution was added, mixed gently to prevent foaming and incubated at room temperature for 15–20 min in the dark. Subsequently, an appropriate volume of propidium iodide (PI) staining solution was added and mixed gently. Cells were then washed gently with $1\times$ PBS to remove un-bound Annexin V-FITC and PI dyes. After staining, the cells were examined under a fluorescent microscope (Nikon Eclipse 800), and the corresponding fluorescent images were captured at a $20\times$ magnification.

To quantify the percentage of Annexin V-positive and PI-positive cells, multiple fields were randomly chosen. The numbers of Annexin V-positive (green fluorescence) and PI-positive (red fluorescence) cells were enumerated for each field, along with the total number of cells using brightfield microscopy. The percentage of Annexin V-positive and PI-positive cells was calculated as the number of positive cells divided by the total number of cells and then multiplied by 100. The results were then averaged across all the fields to determine the overall percentage of Annexin V-positive and PI-positive cells for each experimental condition.

### 2.10. Statistical Analysis

All experimental results were analyzed using Student's *t*-tests. A *p*-value less than 0.05 was considered statistically significant.

## 3. Results

### 3.1. AKR1B1 Expression in Human Glioma Tissue and Various Glioma Cell Lines

To explore whether AKR1B1 expression was inhibited in human glioma cells, we examined AKR1B1 mRNA levels in normal human brain tissues and glioma tissues. According to the analysis by quantitative reverse-transcription polymerase chain reaction, the AKR1B1 mRNA level significantly decreased by 50% in human glioma tissues (Figure 1A), suggesting that low AKR1B1 levels participated in tumorigenesis.

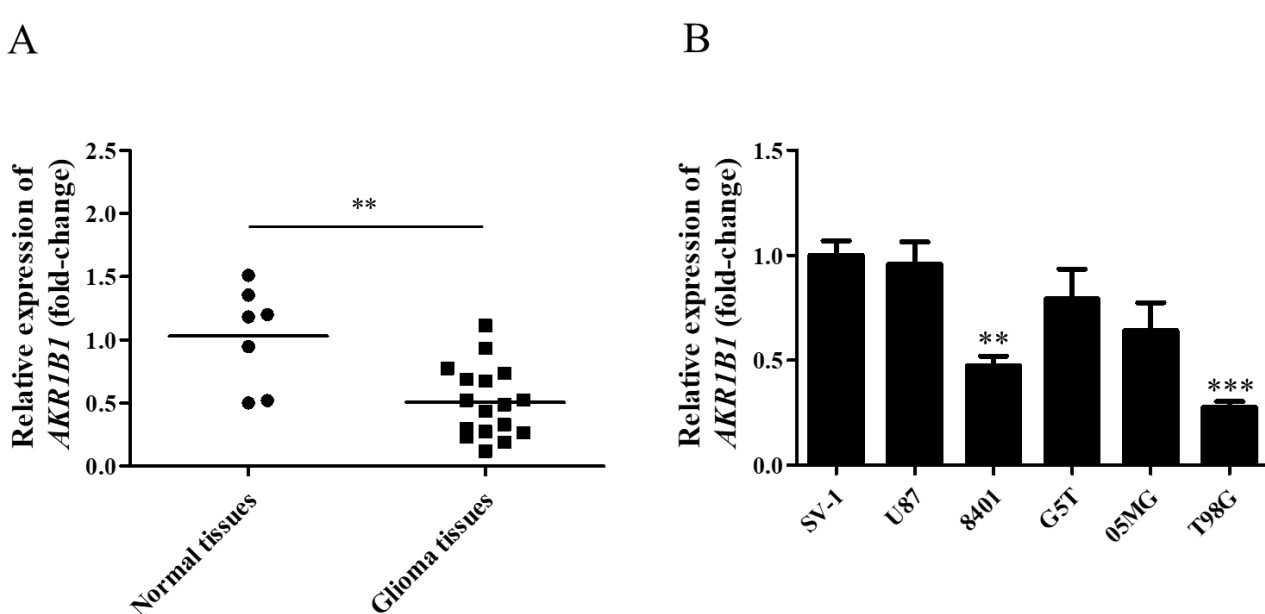

**Figure 1.** AKR1B1 mRNA levels in human glioma tissue and cell lines. (**A**) AKR1B1 mRNA expression in glioma (squares) and normal brain (dots) tissues. (**B**) Comparison of AKR1B1 mRNA levels between various glioma cell lines and SV-1 cells (*n* = 3/cell type). Data are presented as mean ± SEM comparing the mean of AKR1B1 expression; ** *p* < 0.01, and *** *p* < 0.001.

According to our findings in human tissues, human glioma cell lines with low AKR1B1 expression should be selected as the appropriate cell model. Among normal human astrocytes (SV-1) and human glioma cell lines (U87, 8401, G5T, 05MG, and T98G), 8401 and T98G cell lines presented the lowest AKR1B1 mRNA levels (Figure 1B). Therefore, 8401 and T98G cells were the most appropriate cell models to investigate the role of AKR1B1 in tumorigenesis.

### 3.2. AKR1B1 Caused a Cytotoxic Effect on Human Glioblastoma Cell Lines

To examine whether AKR1B1 could inhibit tumor growth, we overexpressed AKR1B1 levels in human glioma cells by transfecting plasmid expression into the cells (denoted as

AKR1B1-expressing glioma cells). The transfection efficiency of the AKR1B1-expressing plasmid was determined by increased mRNA and protein levels in human T98G cells (Figure 2A). After plasmid transfection, we examined the cytotoxic effect every 24 h by using the MTT assay. AKR1B1 showed potently antiproliferative effects on T98G and 8401 time-dependently (Figure 3A,B). We further examined whether inhibiting AKR1B1 could promote glioma cell proliferation. To test our hypothesis, we knocked down AKR1B1 in glioma cells through si-RNA and measured cell viability every 24 h. Diminished mRNA and protein levels in 8401 cells determined the transfection efficiency of AKR1B1 si-RNA (Figure 2B). According to the cell viability analysis, AKR1B1 knockdown exacerbated cancer cell growth time-dependently (Figure 3C,D). Therefore, low AKR1B1 levels directly caused tumor progression.

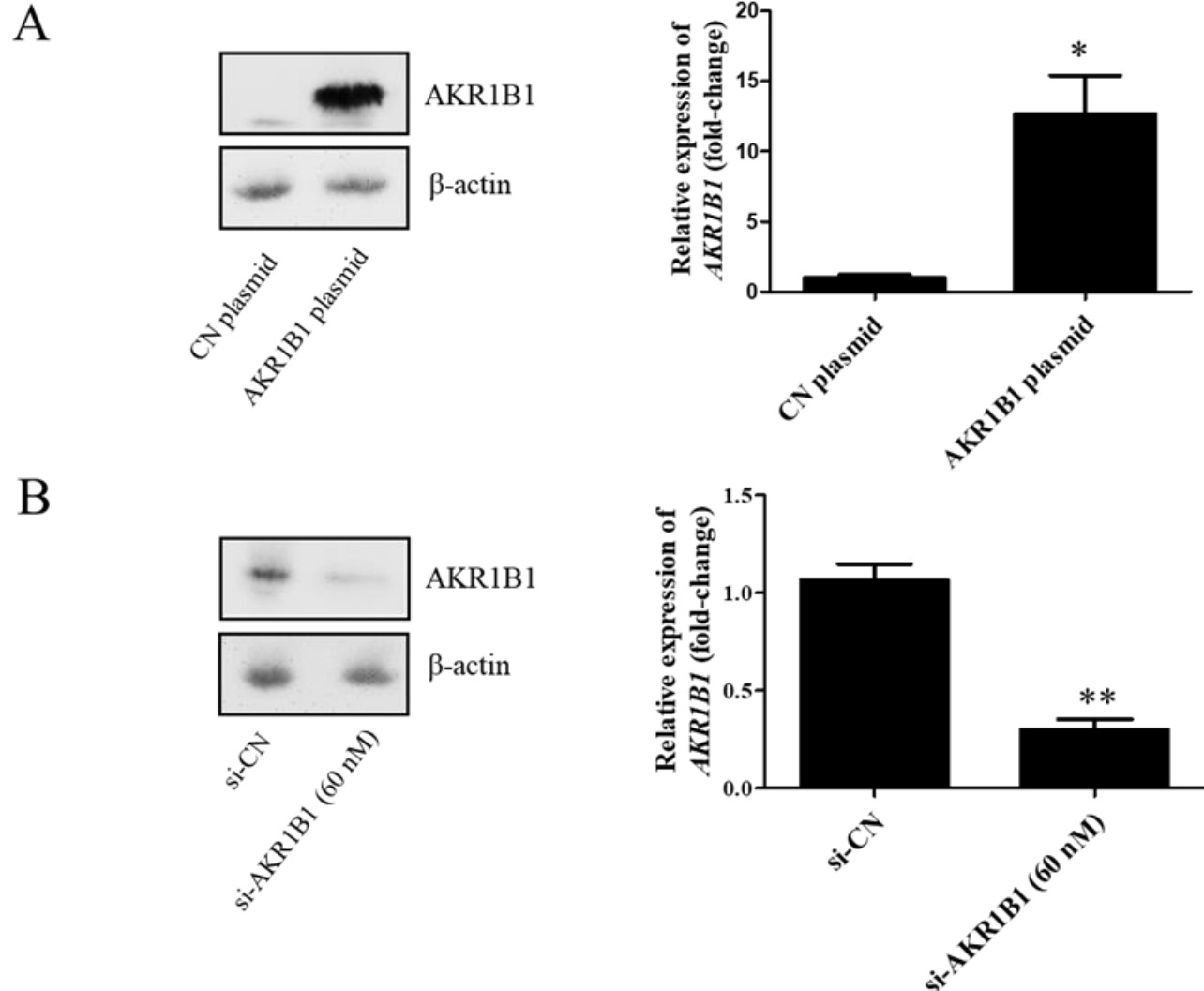

**Figure 2.** Transfection efficiency of the AKR1B1-expressing plasmid and AKR1B1 si-RNA. (**A**) Western blot (left panel) and quantitative reverse-transcription polymerase chain reaction (q-RT-PCR) (right panel) analysis results show increased AKR1B1 protein and mRNA levels at 24 h after transfection with the AKR1B1 plasmid in T98G cells, respectively, ($n = 3$). (**B**) Western blot (left panel) and q-RT-PCR (right panel) analysis results reveal that AKR1B1 protein and mRNA levels significantly declined at 72 h after siRNA transfection in 8401 cells ($n = 3$). Data are presented as mean ± SEM comparing 8401 glioma cells treated with the control plasmid (CN plasmid) or the control si-RNA (si-CN); * $p < 0.05$ and ** $p < 0.01$.

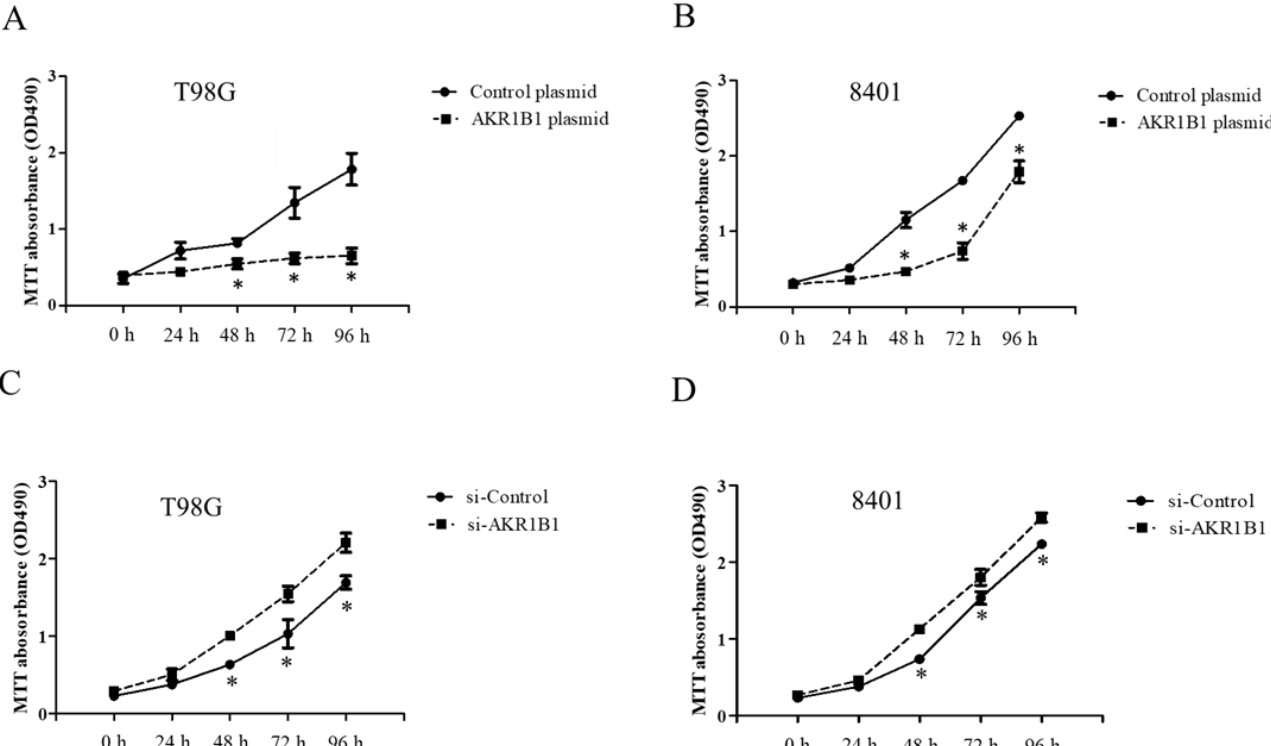

**Figure 3.** Antitumor effect of ARK1B1 on human glioma cells. (**A**,**B**) AKR1B1 reduced the cell viability of human glioma cells. T98G cells (**A**) and 8401 cells (**B**) were transfected with the AKR1B1-expressing plasmid. Cell viability was determined by the MTT assay at 0, 24, 48, 72, and 96 h after plasmid transfection ($n = 3$/cell type). (**C**,**D**) AKR1B1 knockdown increased the cell viability of human glioma cells. T98G cells (**C**) and 8401 cells (**D**) were transfected with AKR1B1 si-RNA, and cell viability was determined by the MTT assay at 0, 24, 48, 72, and 96 h after si-RNA transfection ($n = 3$/cell type). Data are presented as mean ± SEM from three independent experiments; * $p < 0.05$.

### 3.3. AKR1B1 Activated p38 Signaling in Glioma Cells

In the mechanism of AKR1B1-induced cancer cell death, AKR1B1 might activate the p38 MAPK signaling pathway in glioma cells. To test this hypothesis, we assessed phosphorylated p38 MAPK (the active form of p38 MAPK) and total p38 MAPK expression levels in glioma cells by Western blot analysis. We observed increased phosphorylated p38 MAPK levels and no significant alteration of total p38 MAPK protein in AKR1B1-expressing T98G cells (Figure 4A,B). Next, we assessed whether the antitumor effect of AKR1B1 could be reversed by SB203580, a p38 MAPK inhibitor. Indeed, the p38 MAPK inhibitor rescued the viability of T98G cells time-dependently (Figure 4B). Thus, AKR1B1 directly activated p38 MAPK signaling, which played a critical role in the antitumor effect of AKR1B1.

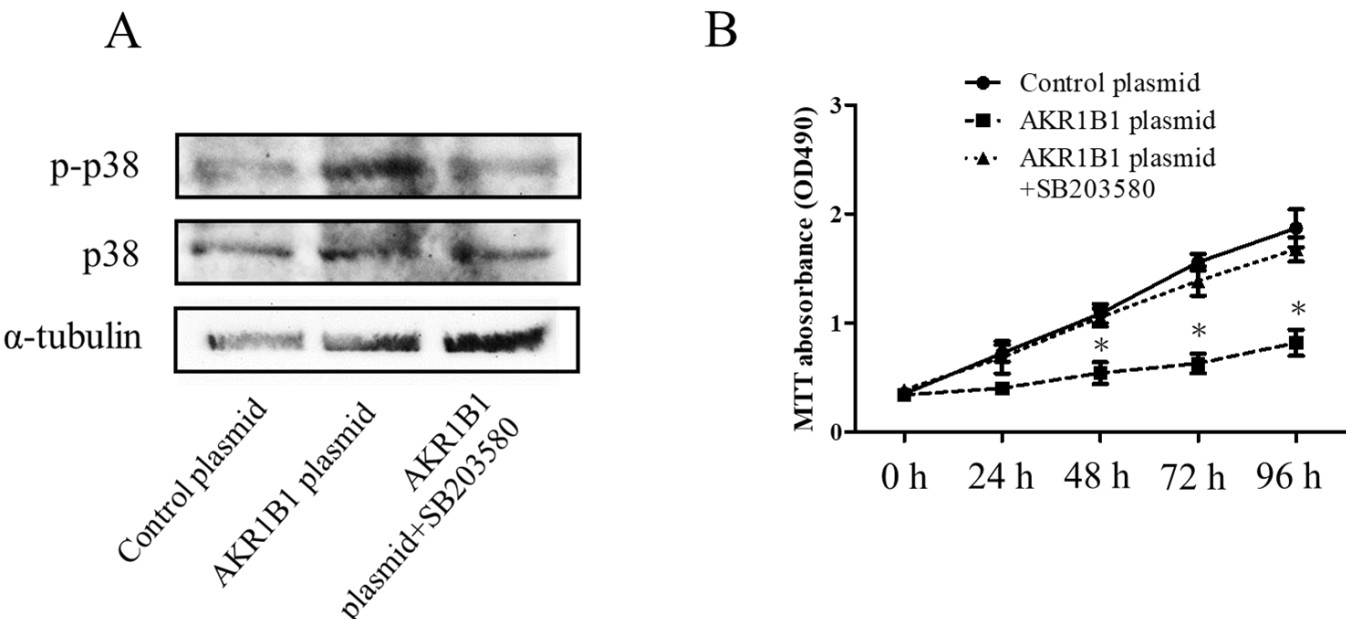

**Figure 4.** AKR1B1 exerted an antitumor effect by regulating the p38 MAPK signaling pathway. (**A**) Representative Western blot results reveal increased p38 MAPK phosphorylation in T98G cells with overexpressed AKR1B1 (*n* = 3). (**B**) Ten micromolar of p38 MAPK inhibitor (SB203580) reversed the antiproliferative effect of AKR1B1 on T98Gcells. T98G cells were transfected with the control plasmid, the AKR1B1 plasmid, and the AKR1B1 plasmid with SB203580. Cell viability was measured by the MTT assay at 0, 24, 48, 72, and 96 h after transfection (*n* = 3). Data are presented as mean $\pm$ SEM from three independent experiments; * $p < 0.05$.

### 3.4. AKR1B1 Induced Apoptosis in Glioma Cells

Given that AKR1B1-expressing glioma cells inhibited cell growth (Figure 3A,B) and induced activation of p38 MAPK signaling (Figure 4A,B), the p38 MAPK pathway has been shown to play a role in inducing apoptosis or programmed cell death in cancer cells [35,36]. AKR1B1 might exert a cytotoxic effect through an apoptotic signal pathway. Therefore, we examined the expression of apoptosis-related proteins such as BAX and Bcl-2 in AKR1B1-expressing glioma cells.

Both the mRNA and protein levels of apoptosis-associated BAX and Bcl-2 signals were examined in AKR1B1-expressing T98G cells. After AKR1B1 plasmid transfection, the mRNA level of the pro-apoptotic BAX signal increased time-dependently (Figure 5A), whereas that of the anti-apoptotic Bcl-2 signal significantly decreased (Figure 5B). In addition, the ratio of BAX/Bcl-2 increased time-dependently in the AKR1B1 group but did not significantly change in the control group (Figure 5C). Regarding the protein expression of BAX and Bcl-2 in AKR1B1-expressing T98G cells, the same effect was observed (Figure 5D).

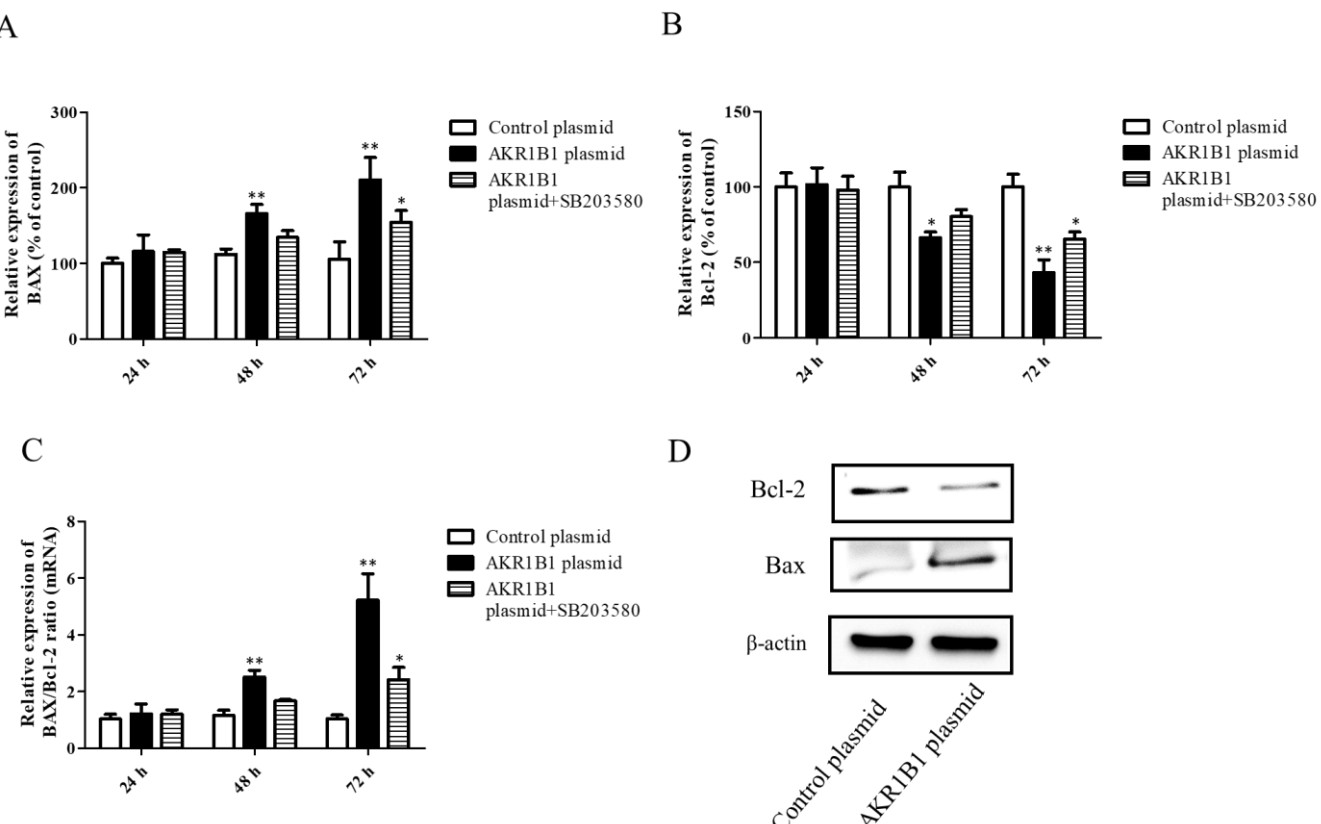

**Figure 5.** AKR1B1 induced cell apoptosis in human glioma cells. (**A–D**) The anti-apoptotic effect was evaluated by assessing the pro-apoptotic BAX signal, the antiapoptotic protein Bcl-2, and the BAX/Bcl-2 ratio in T98G cells with overexpressed AKR1B1. The mRNA levels of BAX (**A**) and Bcl-2 (**B**) in T98G cells transfected with the control, the AKR1B1 plasmid, or SB203580 were measured by quantitative reverse-transcription polymerase chain reaction at 24, 48, and 72 h after transfection. (**C**) The BAX/Bcl-2 ratio increased time-dependently in T98G cells with overexpressed AKR1B1. (**D**) Apoptosis-associated proteins such as BAX and Bcl-2 were measured using Western blot analysis. Data are presented as mean $\pm$ SEM from three independent experiments; * $p < 0.05$ and ** $p < 0.01$.

### 3.5. AKR1B1 Activated Caspase-3/7 in Glioma Cells

Considering that caspase-3/7 plays a major role in cell apoptosis [46–49], we next examined whether caspase-3/7 could be activated in AKR1B1-expressing glioma cells. We transfected the T98G cells with the control plasmid or 0.05/0.1/0.2 µg of the AKR1B1-expressing plasmid and then measured caspase-3/7 activity at 24 and 48 h after the transfection. The caspase-3/7 activity elevated, as the dose of AKR1B1 plasmid increased (Figure 6A,B). The induction of apoptotic cell death by AKR1B1 expression was examined using the Annexin V/PI assay. As depicted in Figure 6C, T98G cells transfected with AKR1B1 expression plasmids for 24 h displayed Annexin V-FITC staining on the surface of apoptotic cells (in green), while PI staining highlighted the nuclei of apoptotic or dead cells (in red). Therefore, apoptosis could be a major contributor to the antitumor effect of AKR1B1 on glioma cells.

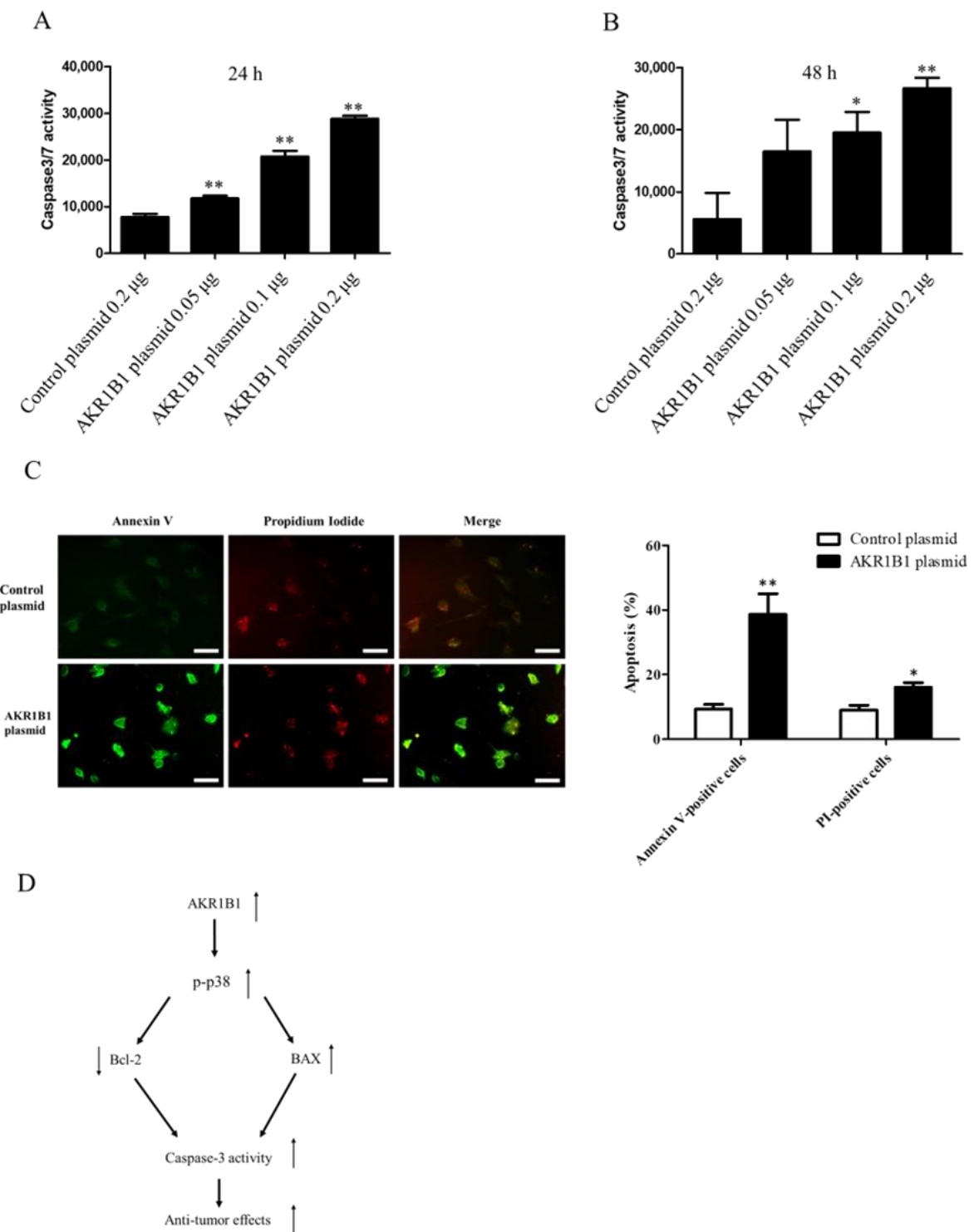

**Figure 6.** AKR1B1 induced cell apoptosis by activating caspase 3/7 activity in glioma cells. (**A**,**B**) AKR1B1 increased caspase3/7 activity in human glioma cells. T98 cells were transfected with the control plasmid or 0.05/0.1/0.2 μg of the AKR1B1-expressing plasmid. Then, the caspase 3/7 activity was measured at 24 h (**A**) and 48 h (**B**) after plasmid transfection. Data are presented as mean ± SEM from three independent experiments; * $p < 0.05$ and ** $p < 0.01$. (**C**) After treatment, cells were subjected to an Annexin-V/PI double staining assay. Specifically, the cells were stained with Annexin V-FITC (green) and PI (red) and subsequently analyzed using a fluorescence microscope. The scale bar represents 100 μm. (**D**) A schematic diagram shows AKR1B1 exerting antitumor effects by activating the apoptotic pathways.

## 4. Discussion

This study demonstrated that AKR1B1 was significantly decreased in glioma tissues compared with that in the adjacent tissues. AKR1B1 also decreased in glioma cell lines (T98G and 8401) in comparison with that in human astrocytes. The MTT assays showed that AKR1B1 overexpression inhibited glioma cell proliferation whereas AKR1B1 knockdown increased it. Nonetheless, AKR1B1-induced p38 MAPK phosphorylation and SB203580 could reverse this inhibitory effect on cell proliferation. Furthermore, AKR1B1 overexpression increased the BAX protein and mRNA expression levels and the Bcl-2/BAX ratio but decreased the Bcl-2 protein and mRNA expression levels. AKR1B1 also induced caspase-3/7 activity.

Yamada et al. found that AKR1B1 is significantly hypermethylated and decreases in hepatocellular carcinoma tumors compared with that in normal liver tissues [9]. In adrenocortical carcinomas and prostate cancer, AKR1B1 expression is decreased, but the mechanism and function remain unknown [6,11]. In the present study, AKR1B1 expression was significantly decreased in glioma tissues compared with that in adjacent normal tissues. The AKR1B1 level was also decreased in glioma cell lines such as T98G and 8401 in comparison with that in human astrocytes (Figure 1A,B). Based on our observations, we speculate that the low expression of AKR1B1 may involve glioma progression.

Moreover, p38 MAPK signaling plays an important role in GBM. The kinase p38 induces apoptosis in GBM cells while inhibiting p38 phosphorylation prevents apoptosis [50]. Earlier research suggests that the activation of p38 MAPK may have a positive impact on certain aspects of glioma therapy. Yao et al. discovered that the anti-proliferative effect of β-elemene on glioblastoma cells was dependent on the activation of p38 MAPK, and the inhibition of p38 MAPK reversed the anti-proliferative effect of β-elemene [51]. Therefore, p38 MAPK may be a potential target for glioma therapy.

Our data showed that AKR1B1 overexpression can induce p38 MAPK phosphorylation and inhibit glioma cell proliferation (Figures 3 and 4). However, the inhibitory effect of AKR1B1 on cell proliferation can be reversed by a p38 MAPK inhibitor (SB203580) (Figure 4B). Therefore, targeting p38 MAPK may be an underlying mechanism by which AKR1B1 inhibits GBM proliferation. Reactive oxygen species (ROS) are potent oxidative agents that can serve as second messengers to directly or indirectly modulate the activation of the p38 MAPK pathway when their intracellular concentrations escalate. The activation of the p38 MAPK pathway augments cellular responses to ROS and participates in regulating apoptosis, inflammation, and other stress responses [52]. Previous investigations have revealed that arsenic trioxide stimulates ROS generation, activates the p38 MAPK signaling pathway and promotes apoptosis in cancer cells [53]. AKR1B1 plays a role in the metabolism and reduction of bioactive aldehydes, which can impact the intracellular redox balance and consequently alter intracellular ROS levels [14]. Therefore, we hypothesize that AKR1B1-induced phosphorylation of p38 MAPK may transpire through an elevation in intracellular ROS concentrations.

AKR1B1 metabolizes glucose into sorbitol via the polyol pathway [54]. Sorbitol has been found to possibly possess antitumor properties. One study demonstrated that sorbitol induces apoptosis in colorectal cancer cells by increasing the phosphorylation of p38 MAPK, upregulating the expression of BAX and cleaved caspase-3 while downregulating the expression of Bcl-2 [55]. Another study showed that sorbitol induces apoptosis in gastric cancer cells by regulating PKC activity [56]. Furthermore, it has been demonstrated that sorbitol can induce p38 MAPK phosphorylation and enhance chemosensitivity in T98G glioma cells [57]. Therefore, AKR1B1-induced p38 MAPK phosphorylation through sorbitol in glioma cells is the focus of our future research.

BAX, a member of the Bcl-2 family, is a core regulator of the intrinsic pathway of apoptosis [58]. Silencing of p38 MAPK by si-RNA-blocked streptococcal pyrogenic exotoxin B induces BAX expression and apoptosis in A549 cells [59]. The treatment of PC12 cells with rotenone significantly induces apoptosis with the p38/p53/BAX signaling axis [37]. Conversely, the anti-apoptotic Bcl-2 proteins inhibit cell death by binding and inhibiting

pro-apoptotic Bcl-2 proteins [58]. In HCT116 and SW480 colorectal cancer cells, inhibiting phosphorylated p38 MAPK reduces the Bcl-2 expression [60]. In our study, AKR1B1 overexpression in T98G cells induced BAX and reduced Bcl-2 expression, but these effects may be regulated by p38 MAPK signaling. In addition, AKR1B1 inhibited T98G cell proliferation and significantly increased the BAX/Bcl-2 ratio (Figure 5).

Caspase-3, which plays an integral role in apoptosis, is a primary target for cancer therapy. Activated caspase-3 stimulates death protease and initiates protein breakdown, leading to apoptosis [49]. The report highlights the high expression of miR-155-5p and miR-221-3p in glioma cells, which inhibit the expression of caspase-3. The use of peptide nucleic acids targeting these two miRNAs has been shown to induce apoptosis in temozolomide-resistant T98G glioma cells by enhancing caspase-3 protein expression [61]. The BAX/Bcl-2 mRNA and protein ratios reportedly correlate with caspase-3 expression [62]. In our study, AKR1B1 overexpression increased the caspase-3/7 activity of glioma cells, suggesting that AKR1B1 is a potential therapeutic effect of gliomas (Figure 6A,B). Caspase-3 plays a pivotal role in cellular apoptosis, and its activity exhibits a positive correlation with the extent of apoptosis. During the initial phase of apoptosis, phosphatidylserine (PS) in the cell membrane undergoes translocation from the inner to the outer leaflet. As apoptosis progresses to the late stage, cells exhibit increased membrane permeability, allowing propidium iodide (PI) to enter and stain the DNA, eventually leading to the disintegration of cellular structures and formation of apoptotic bodies [44]. Our findings demonstrate that AKR1B1 significantly promotes apoptosis in T98G cells, as evidenced by the elevated percentages of Annexin V-positive and PI-positive cells observed in the Annexin V-FITC/PI double-staining assay (Figure 6C). This observation substantiates that AKR1B1 augments caspase-3 activity and initiates downstream pathways.

In conclusion, AKR1B1 has an antitumor effect on glioma cells by inducing the phosphorylated levels of p38 MAPK and thereby increasing the BAX/Bcl-2 ratio and caspase-3/7 activity. Therefore, AKR1B1 may be a promising candidate for glioma treatment.

**Author Contributions:** Conceptualization, Y.-K.H. and C.-L.L.; writing—original draft preparation, Y.-K.H.; methodology and interpretation, K.-C.C. and Y.-K.H.; writing—review and editing, K.-C.C. and C.-Y.L.; formal analysis, A.-S.L.; supervision, C.-L.L.; project administration, C.-L.L. All authors have read and agreed to the published version of the manuscript.

**Funding:** This work was supported by the funding from Kaohsiung Municipal Ta Tung Hospital (kmtth-107-010 and kmtth-108-020), NIH Core Grant P30-EY008098, Eye and Ear Foundation of Pittsburgh, and an unrestricted grant from Research to Prevent Blindness, New York, NY.

**Institutional Review Board Statement:** The study was carried out in compliance with the approval of the Clinical Research Ethics Committee of the Affiliated Hospital of Kaohsiung Medical University (KMUHIRB-G(II)-20170010).

**Informed Consent Statement:** All subjects involved in the study provided informed consent. The patients provided written informed consent for publication of this paper.

**Data Availability Statement:** Not applicable.

**Conflicts of Interest:** The authors declare no conflict of interest.

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
