# Peer review of "AKR1B1 Represses Glioma Cell Proliferation through p38 MAPK-Mediated Bcl-2/BAX/Caspase-3 Apoptotic Signaling Pathways"

_cimb, doi:10.3390/cimb45040222_

Round 1
Reviewer 1 Report
Comments and Suggestions for Authors
In this study by Huang et. al. showed that effect of AKR1B1 expression on glioma cell proliferation via MAPK-mediated Bcl-2/BAX/caspase-3 signaling pathway. The study has been carefully performed and it will be very helpful for future studies in the field. Nonetheless, the authors would need to deal with the following points:
1- What is the normal brain tissue in Fig 1A? Which part of brain? Was this tissue taken from a living person or from a dead person? Please give more details for these questions in manuscript.
2- In figure 1B, SV1 cells have also 1,5 fold AKR1B1 expression, What is your normalized control?
3- Researchers transfected plasmids with reagents to glioma cells, and they did not antibiotic selection (because they collect samples after 24 h for western blot). Was this plasmid high expression plasmid, 24 h can be early? Please write the plasmid name. If there is a GFP, please show the expression levels by microscopy images.
4- For fig 4A, please show the decrease of p-p38 with inhibitor treatment.
5- Does AKR1B1 overexpression increase Bax by blocking Bcl-2, or does it affect both? What is the expression levels of Bcl-2 and Bax in p-p38 inhibition?
6- Apoptotic cell death should be detailed with more experiments, like c-PARP, images or cell viability assays etc.
Author Response
Reviewers' comments:
Review1
Comments and Suggestions for Authors
In this study by Huang et. al. showed that effect of AKR1B1 expression on glioma cell proliferation via MAPK-mediated Bcl-2/BAX/caspase-3 signaling pathway. The study has been carefully performed and it will be very helpful for future studies in the field. Nonetheless, the authors would need to deal with the following points:
1- What is the normal brain tissue in Fig 1A? Which part of brain? Was this tissue taken from a living person or from a dead person? Please give more details for these questions in manuscript.
Answer:
Normal brain tissue was collected during the resection of glioma tissue. As part of the standard procedure for glioma resection, corticotomy was performed, and some normal brain tissue was inadvertently resected. Thus, we were able to collect normal brain tissue during the necessary steps of the tumor resection procedure. The location of the collected normal brain tissue was determined by its proximity to the tumor, such as the frontal, parietal, or occipital lobes.
We have added this information to Section 2.2, "Patient Samples," in the Materials and Methods section of the manuscript.
2- In figure 1B, SV1 cells have also 1,5 fold AKR1B1 expression, What is your normalized control?
Answer: Thank you for your inquiry. In Figure 1B, the signals were normalized to the AKR1B1 expression level in 05MG cells. Therefore, SV1 cells showed a 1.5-fold increase in AKR1B1 expression relative to 05MG cells. However, to provide a more comprehensive representation of AKR1B1 expression levels in various cell lines, we reanalyzed the data using SV-1 cells as the normalized control. We have included a detailed explanation of this in the Figure Legends (Figure 1B) to ensure clarity for readers.
3- Researchers transfected plasmids with reagents to glioma cells, and they did not antibiotic selection (because they collect samples after 24 h for western blot). Was this plasmid high expression plasmid, 24 h can be early? Please write the plasmid name. If there is a GFP, please show the expression levels by microscopy images.
Answer: We utilized the pCMV6-Myc-DDK-tagged-AKR1B1 plasma (OriGene, Rockville, MD, USA) for our study, while the control plasma (OriGene, Rockville, MD, USA) was the pCMV6-Entry vector. Notably, neither of these plasmids was GFP-tagged. In accordance with the suggestion of the reviewer, we have now included the names of these plasmas in the 2.4. DNA transfection and RNA interference of Materials and Methods section.
4- For fig 4A, please show the decrease of p-p38 with inhibitor treatment.
Answer: We express our gratitude to the reviewer for the suggestion. In response to the reviewer's comments, we conducted a Western blot analysis to investigate the impact of the p38 MAPK inhibitor SB203580 on AKR1B1-induced p38 ortho-phosphorylation. The results of this analysis have been included in Figure 4 of the manuscript.
5- Does AKR1B1 overexpression increase Bax by blocking Bcl-2, or does it affect both? What is the expression levels of Bcl-2 and Bax in p-p38 inhibition?
Answer:
- AKR1B1 may impact the expression of Bax and Bcl-2 by elevating the levels of ROS and sorbitol within cells. Our literature review indicates that ROS can increase Bax expression by inhibiting Bcl-2 (Sablina et al., 2005) and activate p53 to induce Bax expression (Sablina et al., 2005). Additionally, sorbitol-induced hyperosmolar stress can stimulate ROS production by activating the MAPK signaling pathway (Johnson & Lapadat, n.d.; Kyriakis & Avruch, 2001). Hence, AKR1B1 overexpression may regulate both of these mechanisms, although further experiments are required to verify this in the future.
- We have conducted experiments to investigate the effect of SB203580 on the expression of Bax and Bcl-2 mRNA induced by AKR1B1, and the results have been presented in Figure 5A-C of the manuscript.
References:
Johnson, G. L., & Lapadat, R. (n.d.). Mitogen-Activated Protein Kinase Pathways Mediated by ERK, JNK, and p38 Protein Kinases. http://stke.sciencemag.org/cgi/cm/stkecm;
Kyriakis, J. M., & Avruch, J. (2001). Mammalian Mitogen-Activated Protein Kinase Signal Transduction Pathways Activated by Stress and Inflammation. http://physrev.physiology.org
Sablina, A. A., Budanov, A. V., Ilyinskaya, G. V., Agapova, L. S., Kravchenko, J. E., & Chumakov, P. M. (2005). The antioxidant function of the p53 tumor suppressor. Nature Medicine, 11(12), 1306–1313. https://doi.org/10.1038/nm1320
6- Apoptotic cell death should be detailed with more experiments, like c-PARP, images or cell viability assays etc.
Answer:
We concur with the reviewer's feedback. Accordingly, we performed Annexin V-FITC/PI double-staining assay to evaluate the induction of apoptosis in T98G cells by AKR1B1, and the results have been presented in Figure 6 C of the manuscript.
Reviewer 2 Report
Comments and Suggestions for Authors
This paper examines the role pf AKR1-p38 MAPK signalling in the apoptosis of Glioma cells. It needs some refinement of the English language. The experiments and interpretations are clear. The general impact medium-low. What is the connection of the AKR enzyme to the p38 kinase ? P38 is regulated by other kinases and phosphatases is not addressed. How AKR regulates these enzymes that regulate p38? Another point is that public data sets from the CGGA data base support an oncogenic rather than a tumor suppressor role that the present manuscripts data indicate. I would recommend resubmission after extensive revisions
[WU1]
Author Response
Review2
Comments and Suggestions for Authors
This paper examines the role pf AKR1-p38 MAPK signalling in the apoptosis of Glioma cells. It needs some refinement of the English language. The experiments and interpretations are clear. The general impact medium-low. What is the connection of the AKR enzyme to the p38 kinase? P38 is regulated by other kinases and phosphatases is not addressed. How AKR regulates these enzymes that regulate p38? Another point is that public data sets from the CGGA data base support an oncogenic rather than a tumor suppressor role that the present manuscripts data indicate. I would recommend resubmission after extensive revisions
Answer:
- AKR1B1 may exert its effect on secondary signal transmission, such as p38, by increasing intracellular levels of ROS and sorbitol (Johnson & Lapadat, n.d.; Kyriakis & Avruch, 2001). Previous studies have demonstrated that elevated intracellular ROS levels can activate MAPK kinases, including MAPKKK and MAPKK, leading to the phosphorylation and activation of p38 (Kamata et al., 2005). Furthermore, hyperosmolar stress induced by sorbitol can augment ROS production via activation of the MAPK signaling pathway (Johnson & Lapadat, n.d.; Kyriakis & Avruch, 2001). Additionally, studies have suggested that sorbitol can increase the level of p38 phosphorylation, thus activating the p38 MAPK pathway (Lu et al., 2014).
- We appreciate the valuable comments from the reviewer. We have taken note of the information from the CGGA database regarding AKR1B1 performance in glioma. However, we would like to clarify that our study used tumor-adjacent normal tissue and glioma grade IV samples, which differ from the data in the CGGA database. We acknowledge that the differential expression of AKR1B1 observed in our study warrants further investigation to elucidate its underlying mechanisms.
- We would like to express our gratitude to the native English-speaking professionals who have assisted us in improving the language quality of our manuscript.
References:
Johnson, G. L., & Lapadat, R. (n.d.). Mitogen-Activated Protein Kinase Pathways Mediated by ERK, JNK, and p38 Protein Kinases. http://stke.sciencemag.org/cgi/cm/stkecm;
Kamata, H., Honda, S. I., Maeda, S., Chang, L., Hirata, H., & Karin, M. (2005). Reactive oxygen species promote TNFα-induced death and sustained JNK activation by inhibiting MAP kinase phosphatases. Cell, 120(5), 649–661. https://doi.org/10.1016/j.cell.2004.12.041
Kyriakis, J. M., & Avruch, J. (2001). Mammalian Mitogen-Activated Protein Kinase Signal Transduction Pathways Activated by Stress and Inflammation. http://physrev.physiology.org
Lu, X., Li, C., Wang, Y. K., Jiang, K., & Gai, X. D. (2014). Sorbitol induces apoptosis of human colorectal cancer cells via p38 MAPK signal transduction. Oncology Letters, 7(6), 1992–1996. https://doi.org/10.3892/ol.2014.1994
Reviewer 3 Report
Comments and Suggestions for Authors
The manuscript “AKR1B1 represses glioma cell proliferation through p38 MAPK-mediated Bcl-2/BAX/caspase-3 apoptotic signaling pathways” represents an original article focusing on a certain enzyme AKR1B1 and selected related molecular pathways in glial tumours. The topic corresponds to the aims and spectrum of the journal. It also fits in the current research directions to identify potential molecular targets for future treatment approaches.
The applied methods are up-to-dated and comprehensively described. The results are explicitly presented and supplemented by appropriate figures. The manuscript is characterised by clear design, logical structure and high level of English language, ensuring good scientific comprehensibility; it is well-written and truly interesting.
Few aspects could be clarified in order to further improve the manuscript:
1) I would strongly suggest to avoid the phrase “normal glioma tissues” (in the Abstract), since tumour tissues are NOT normal.
2) Regarding downregulation of AKR1B1 in human glioma tissues, how many patient samples were evaluated?
3) Please, explain, what kind of human gliomas were studied? Is the term “glioma” used as a synonym of glioblastoma? If yes, explain this in the manuscript, please. If no, please, characterise the tested gliomas by the spectrum of diagnoses and grades according to the current WHO classification (2021).
4) Glioblastomas are classified into several molecular types. Is the downregulation of AKR1B1 evident in all molecular types? If this aspect is beyond your study, was the downregulation of AKR1B1 homogeneous, i.e., present in all the tested samples of human glioma?
5) In the subsection “Materials and methods”, it is noted that written informed consent has been obtained from each participant. However, at the end of manuscript, it is stated that informed consent statement is not applicable to the given study. Could you kindly clarify, please?
6) Similarly, in “Materials and methods”, it is noted that the study was approved by Clinical Research Ethics Committee of Kaohsiung Medical University Hospital. At the end of manuscript, authors write that institutional review board statement is not applicable to the given study. Could you kindly clarify this controversy, please?
7) Although the article is outstanding from the scientific point of view, some formatting issues should be solved, e.g., formatting of some references (Ref.9, 10, 13, among others).
Finally, I would like to thank the authors for their huge work input. It was a true honour and a great pleasure to review this manuscript.
Author Response
Review3
Comments and Suggestions for Authors
The manuscript “AKR1B1 represses glioma cell proliferation through p38 MAPK-mediated Bcl-2/BAX/caspase-3 apoptotic signaling pathways” represents an original article focusing on a certain enzyme AKR1B1 and selected related molecular pathways in glial tumours. The topic corresponds to the aims and spectrum of the journal. It also fits in the current research directions to identify potential molecular targets for future treatment approaches.
The applied methods are up-to-dated and comprehensively described. The results are explicitly presented and supplemented by appropriate figures. The manuscript is characterised by clear design, logical structure and high level of English language, ensuring good scientific comprehensibility; it is well-written and truly interesting.
Few aspects could be clarified in order to further improve the manuscript:
1) I would strongly suggest to avoid the phrase “normal glioma tissues” (in the Abstract), since tumor tissues are NOT normal.
Answer:
We appreciate the valuable suggestion from the reviewer. As recommended, we have modified the sentence in the abstract to replace "normal glioma tissues" with "glioma tissues". Thank you for helping us improve the clarity and accuracy of our manuscript.
2) Regarding downregulation of AKR1B1 in human glioma tissues, how many patient samples were evaluated?
Answer:
We assessed 17 samples from patients.
3) Please, explain, what kind of human gliomas were studied? Is the term “glioma” used as a synonym of glioblastoma? If yes, explain this in the manuscript, please. If no, please, characterise the tested gliomas by the spectrum of diagnoses and grades according to the current WHO classification (2021).
Answer:
We appreciate the valuable comments and suggestions provided by the reviewer. To clarify, in our manuscript, the term glioma is used interchangeably with glioblastoma to refer specifically to grade IV gliomas. This is in line with common clinical practice, where glioma is often used as a synonym for glioblastoma. To address this, we will include the following sentence in the first paragraph of the introduction: "In this manuscript, the term glioma is used as a synonym for glioblastoma, referring specifically to grade IV gliomas as per the WHO classification."
The content is as follows:
“Gliomas are a collection of brain tumors that arise from glial cells in the central nerv-ous system (CNS). Glioblastoma, also referred to as glioblastoma multiforme (GBM), represents the most aggressive and malignant subtype of gliomas, classified as grade IV glioma according to the World Health Organization (WHO) classification system. In this manuscript, the terms glioma and glioblastoma are used interchangeably to spe-cifically denote grade IV gliomas, characterized by swift growth, high invasiveness, and unfavorable prognosis.”
4) Glioblastomas are classified into several molecular types. Is the downregulation of AKR1B1 evident in all molecular types? If this aspect is beyond your study, was the downregulation of AKR1B1 homogeneous, i.e., present in all the tested samples of human glioma?
Answer:
Our samples consist entirely of GBM specimens. Out of the 17 GBM samples analyzed, 2 samples exhibited high levels of AKR1B1 content (close to or exceeding the average value of normal tissues), while the remaining samples exhibited uniform levels.
5) In the subsection “Materials and methods”, it is noted that written informed consent has been obtained from each participant. However, at the end of manuscript, it is stated that informed consent statement is not applicable to the given study. Could you kindly clarify, please?
Answer:
We appreciate the reviewer's inquiry. We have made the necessary corrections to the manuscript. The revised statement is as follows:
Informed Consent Statement: All subjects involved in the study provided informed consent. The patients provided written informed consent for publication of this paper.
6) Similarly, in “Materials and methods”, it is noted that the study was approved by Clinical Research Ethics Committee of Kaohsiung Medical University Hospital. At the end of manuscript, authors write that institutional review board statement is not applicable to the given study. Could you kindly clarify this controversy, please?
Answer:
We appreciate the valuable suggestions from the reviewers. We have revised the manuscript as follows:
Institutional Review Board Statement: The study was carried out in compliance with the approval of the Clinical Research Ethics Committee of the Affiliated Hospital of Kaohsiung Medical University (KMUHIRB-G(II)-20170010).
7) Although the article is outstanding from the scientific point of view, some formatting issues should be solved, e.g., formatting of some references (Ref.9, 10, 13, among others). Finally, I would like to thank the authors for their huge work input. It was a true honour and a great pleasure to review this manuscript. We have corrected this sentence.
Answer:
We have conducted a thorough review of the manuscript.
Reviewer 4 Report
Comments and Suggestions for Authors
The article presented by Huang Y. et al. aimed to investigate the role of Aldo‐AKR1B1 in glioma cell proliferation through p38 MAPK activation to further control Bcl-2/BAX/caspase-3 apoptosis signaling. The topic is exciting; however, several aspects need to be profoundly revised and additional experiments are required to validate principal findings.
1. AKR1B1 has been postulated as a prognostic biomarker in malignant gliomas (doi: 10.1155/2022/9979200). Where the expression of AKR1B1 was significantly elevated in glioma tissues compared to normal tissues and the high expression of AKR1B1 was significantly associated with WHO grade. In this sense, which scenario is representing in this study? This must be explained in order to confirm the proposed hypothesis.
2. Please provide a full description of brain tumor tissue samples and normal brain tissues evaluated.
3. AKR1B1 overexpression is not a valid methodology to investigate its cytotoxicity effect. High levels of any protein achieved by transfection can be harmful to cells and ultimately do not represent real scenarios.
4. To whom the AKR1B1 mRNA levels in multiple glioma cell lines were relativized. It is not explained in Figure 1.
5. Why MTT results were measured in 490 nm wavelength? It is not the proper wavelength for violet formazan.
6. The connection between AKR1B1 and p38 MAPK must be validated with other molecular approach. Probably using the chemical inhibitors.
7. Once again, In gliomas, phosphorylated p38 MAPK has been considered a potential biomarker of progression once its activation contributes to tumor invasion and metastasis and is positively correlated with the tumor grade (doi: 10.3389/fphar.2022.975197). In this sense, what possible scenario is being evaluated in this article?
8. Without these clarifications of the type of glioma that is intended to be evaluated, complement the analysis and interpretation tools that were used, the results lack real interpretation.
Author Response
Review4
The article presented by Huang Y. et al. aimed to investigate the role of Aldo‐AKR1B1 in glioma cell proliferation through p38 MAPK activation to further control Bcl-2/BAX/caspase-3 apoptosis signaling. The topic is exciting; however, several aspects need to be profoundly revised and additional experiments are required to validate principal findings.
- AKR1B1 has been postulated as a prognostic biomarker in malignant gliomas (doi: 10.1155/2022/9979200). Where the expression of AKR1B1 was significantly elevated in glioma tissues compared to normal tissues and the high expression of AKR1B1 was significantly associated with WHO grade. In this sense, which scenario is representing in this study? This must be explained in order to confirm the proposed hypothesis.
Answer:
In this study, we investigated the expression of AKR1B1 in 7 normal brain tissues and 17 GBM tissues and observed a significant decrease in AKR1B1 expression in malignant glioma. Moreover, we found that high AKR1B1 expression in glioma cells induces apoptosis, suggesting a potential role for AKR1B1 in the regulation of apoptosis in malignant glioma.
While our findings differ from some previous reports of high AKR1B1 expression in malignant glioma, our glioma cell data support the involvement of AKR1B1 in apoptosis regulation. These differences may be attributed to variations in sample size, source, and experimental methods.
However, our study has some limitations, including the small sample size and tissue samples from different sources. Future investigations should include larger sample sizes and multiple experimental approaches to further explore the function of AKR1B1 in malignant glioma. Furthermore, comparing different types of gliomas may reveal the specific role of AKR1B1 in these tumors and its relationship with apoptosis regulation.
- Please provide a full description of brain tumor tissue samples and normal brain tissues evaluated.
Answer:
During the resection of glioma tissue, normal brain tissue was inadvertently collected as part of the standard procedure for glioma resection. Corticotomy was performed during the procedure, allowing for the collection of normal brain tissue during the necessary steps of the tumor resection procedure. The location of the collected normal brain tissue was determined based on its proximity to the tumor, such as the frontal, parietal, or occipital lobes.
- AKR1B1 overexpression is not a valid methodology to investigate its cytotoxicity effect. High levels of any protein achieved by transfection can be harmful to cells and ultimately do not represent real scenarios.
Answer:
We appreciate the reviewer's comment and have taken it into consideration. As such, we included the pCMV6-Entry vector as a control group in our experiment design. This was done to ensure that if the plasmid caused cytotoxicity in cells, the control group would also be observed under the same conditions. Additionally, we utilized siRNA to decrease AKR1B1 expression and monitor cell growth, as demonstrated in Figure 3 C and D.
- To whom the AKR1B1 mRNA levels in multiple glioma cell lines were relativized. It is not explained in Figure 1.
Answer:
Thank you for your inquiry. In Figure 1B, the signal was normalized to AKR1B1 expression levels in 05MG cells. However, to provide a more comprehensive representation of AKR1B1 expression levels across different cell lines, we reanalyzed the data using SV-1 cells as a normalization control. We have explained this in detail in the figure legend (Figure 1B) to ensure clear understanding for the reader.
- Why MTT results were measured in 490 nm wavelength? It is not the proper wavelength for violet formazan.
Answer:
Although the maximum absorption wavelength of formazan is around 570nm, the commonly used measurement wavelength is 490nm. This is due to the fact that at 490nm, formazan absorption is still high while minimizing background interference from organelles and cell membranes, which can cause interference at 570nm. Additionally, measurements at 490nm are well-correlated with those at 570nm.
The following is a reference that also uses a wavelength of 490nm for MTT assay: MicroRNA-181a sensitizes human malignant glioma U87MG cells to radiation by targeting Bcl-2. Oncology Reports, 23(4). https://doi.org/10.3892/or_00000725
- The connection between AKR1B1 and p38 MAPK must be validated with other molecular approach. Probably using the chemical inhibitors.
Answer:
We greatly appreciate the valuable suggestion from the reviewer. As demonstrated in Figure 4, treatment with the p38 MAPK inhibitor (SB203580) reversed AKR1B1-induced inhibition of cell growth. Additionally, in Figure 5, we observed that SB203580 reversed AKR1B1-induced BAX expression and inhibited Bcl-2 expression. In future studies, we plan to employ additional p38 MAPK inhibitors and siRNA to further investigate the effects of AKR1B1 on cellular physiological processes.
- Once again, In gliomas, phosphorylated p38 MAPK has been considered a potential biomarker of progression once its activation contributes to tumor invasion and metastasis and is positively correlated with the tumor grade (doi: 10.3389/fphar.2022.975197). In this sense, what possible scenario is being evaluated in this article?
Answer:
In our study, we aimed to investigate the impact of AKR1B1 on the p38 MAPK signaling pathway and its potential role in glioma cells. Through our experiments, we explored the regulation of p38 MAPK phosphorylation by AKR1B1 and its related biological effects.
Specifically, our study evaluated the influence of AKR1B1 on the growth and apoptosis of glioma cells by regulating the p38 MAPK signaling pathway. Our results demonstrated a significant impact of AKR1B1 on p38 MAPK phosphorylation, which consequently influenced the biological behaviors of glioma cells, including proliferation and apoptosis.
Our findings align with the existing references on the relationship between phosphorylated p38 MAPK and glioma cells progression in glioma cells (Amantini et al., 2007; Li et al., 2020; Wang et al., 2013). However, our research primarily focuses on the regulatory mechanisms of AKR1B1 on the p38 MAPK signaling pathway, in order to identify new potential targets for glioma therapy.
Future studies should aim to provide further insights into the regulation of the p38 MAPK signaling pathway by AKR1B1, including direct or indirect effects, and other potential upstream and downstream regulators. Additionally, exploring the relationship between AKR1B1 and p38 MAPK signaling pathways in different types and grades of gliomas would be of great interest.
References:
Amantini, C., Mosca, M., Nabissi, M., Lucciarini, R., Caprodossi, S., Arcella, A., Giangaspero, F., & Santoni, G. (2007). Capsaicin-induced apoptosis of glioma cells is mediated by TRPV1 vanilloid receptor and requires p38 MAPK activation. Journal of Neurochemistry, 102(3), 977–990. https://doi.org/10.1111/j.1471-4159.2007.04582.x
Li, Q., Miao, Z., Wang, R., Yang, J., & Zhang, D. (2020). Hesperetin Induces Apoptosis in Human Glioblastoma Cells via p38 MAPK Activation. Nutrition and Cancer, 72(3), 538–545. https://doi.org/10.1080/01635581.2019.1638424
Wang, Z. S., Luo, P., Dai, S. H., Liu, Z. Bin, Zheng, X. R., & Chen, T. (2013). Salvianolic acid b induces apoptosis in human glioma U87 cells through p38-mediated ROS generation. Cellular and Molecular Neurobiology, 33(7), 921–928. https://doi.org/10.1007/s10571-013-9958-z
- Without these clarifications of the type of glioma that is intended to be evaluated, complement the analysis and interpretation tools that were used, the results lack real interpretation.
Answer:
Thank you for your insightful comments. We recognize the importance of clarifying the glioma types and further elaborating on the analysis and interpretation tools used in our study.
In our research, we focused on a specific subset of gliomas, namely glioblastomas (GBMs), which were well represented in our experiments and data analyses. We carefully designed our experimental approach and sample selection to ensure comprehensive coverage of this specific type of glioma.
We employed a diverse range of experimental and computational methods, including cell experiments, signaling pathway analysis, and gene expression analysis, as detailed in the manuscript, to fully elucidate our analytical approach.
Nevertheless, we acknowledge the need to further clarify the glioma types and the analysis and interpretation tools employed, which we will address in the revised manuscript to ensure that the results are more realistic and interpretable. Additionally, we will emphasize the relevance of our findings to the existing literature in the Discussion section to enhance the overall quality of the article.
Thank you again for your valuable feedback. We will strive to improve our manuscript to meet the high standards of the journal.
Round 2
Reviewer 1 Report
Comments and Suggestions for Authors
Appreciate the sincere efforts of the authors to provide thorough and satisfactory answers to my earlier comments. As a result, the paper is much improved and contains compelling data.
This reviewer has no further concerns.
Author Response
Reviewer 1:
Appreciate the sincere efforts of the authors to provide thorough and satisfactory answers to my earlier comments. As a result, the paper is much improved and contains compelling data. This reviewer has no further concerns.
ANS:
Thank you for your positive feedback and recognition of our efforts to address your concerns. We appreciate your valuable comments, which have helped us improve the quality of our paper. We are pleased to know that you are satisfied with the revisions and have no further concerns.
Reviewer 4 Report
Comments and Suggestions for Authors
The article has been considerably improved. Most of my comments have been taken into account. I don't think the Annexin V/PI assay has been properly evaluated. The images do not show what the authors comment, another type of analysis such as flow cytometry would help the interpretation of that particular assay.
Author Response
Reviewer 4:
The article has been considerably improved. Most of my comments have been taken into account. I don't think the Annexin V/PI assay has been properly evaluated. The images do not show what the authors comment, another type of analysis such as flow cytometry would help the interpretation of that particular assay.
ANS:
We appreciate the reviewer's suggestion of using flow cytometry analysis to further support our findings. However, due to experimental limitations, we are unable to perform flow cytometry analysis in our study. To address the reviewer's concerns, we have adopted an alternative approach to strengthen our data interpretation and provide a more robust analysis of the Annexin V/PI assay using fluorescence microscopy.
We have conducted three independent experiments and analyzed three randomly selected fields from each experiment to calculate the total cell count and the number of Annexin V or PI-positive cells. This approach allowed us to obtain quantitative data presented as percentages, which we used to further reinforce the relationship between our findings and apoptosis (page 14, second to last paragraph of the discussion).
Additionally, we have revised Figure 6 (page 12) to include the quantitative data and updated the Materials and Methods section (2.9. Annexin V-FITC/PI double-staining assay, page 5) to provide a more detailed description of our approach to obtain quantitative data from the assay.
We hope that these improvements will address the reviewer's concerns and provide a more comprehensive interpretation of our findings, despite the unavailability of flow cytometry data.
Round 3
Reviewer 4 Report
Comments and Suggestions for Authors
I have no more comments.